

# Multiscale Roughness Influence on Conservative Solute Transport in Self-affine Fractures

*Zhi Dou[a,b], Brent Sleep[b], Hongbin Zhan[c], Zhifang Zhou[a], and Jinguo Wang[a]*

*[a] School of Earth Sciences and Engineering, Hohai University, Nanjing, 210098 China*

*[b] Department of Civil Engineering, University of Toronto, 35 St. George Street, Toronto, ON M5S 1A4, Canada*

*[c] Department of Geology and Geophysics, Texas A&M University, College Station, TX 77843-3115, USA*

*Correspondence to: Zhi Dou (douz@hhu.edu.cn); Brent Sleep (Sleep@ecf.utoronto.ca )*

**Key Points:**

- Investigating the influence of large and small-scale roughness on transport of a conservative solute through a self-affine fracture.

- Removing small-scale secondary roughness delayed the arrival time and shortened the tail of BTCs.

- Small-scale secondary roughness decreased MIM mobile domain fraction and increased mass exchange rate between immobile and mobile domains.





**Abstract:** In this article, the influence of multiscale roughness on transport of a conservative solute through a self-affine

fracture was investigated. The fracture roughness was decomposed into two different scales (i.e., a small-scale stationary

secondary roughness superimposed on a large-scale non-stationary primary roughness) by a wavelet analysis technique.

The fluid flow in the single fracture was characterized by Forchheimer's law and exhibited nonlinear flow features such

as eddies and tortuous streamlines. The results indicated that the small-scale secondary roughness was primarily

responsible for the nonlinear flow features. Numerical simulations of asymptotic conservative solute transport showed

non-Fickian transport characteristics (i.e., early arrivals and long tails) in breakthrough curves (BTCs) and in residence

time distributions (RTDs) with and without consideration of the secondary roughness. Analysis of multiscale BTCs and

RTDs showed that the small-scale secondary roughness played a significant role in enhancing the non-Fickian transport

characteristics. Removing small-scale secondary roughness delayed the arrival time and shortened the tail. The peak

concentrations in BTCs decreased as the secondary roughness was removed, implying that the secondary roughness

could also enhance the solute dilution. Fitting the one-dimensional (1D) Fickian advection-dispersion equation (ADE) to

the numerical BTCs resulted in considerable errors that decreased with the small-scale secondary roughness being

removed. The 1D mobile-immobile model (MIM) provided a better fit to the numerical BTCs and inclusion of the small-

scale secondary roughness in numerical simulations resulted in a decreasing MIM mobile domain fraction and an

increasing mass exchange rate between immobile and mobile domains.

*Keywords*: simulation; fracture; self-affine; multiscale roughness; non-Fickian transport; mobile-immobile model.





## 1. Introduction

Characterizing the transport behavior of a conservative solute through a heterogeneous geological formation is crucial
for many environmental and hydrogeological problems such as enhanced oil recovery, geothermal energy development,
remediation of contaminated groundwater, and carbon storage. The heterogeneity of geological formations is ubiquitous
and occurs at all scales in subsurface. Even in a single fracture, the roughness of fracture walls plays an important role in
the transport behavior of a conservative solute at different scales, with solute transport varying from Fickian to non-
Fickian (anomalous) regimes (Dentz et al., 2004). Improving the fundamental understanding and predictability of
conservative solute transport behavior through single fractures at various scales is important for analyzing anomalous
hydrogeological phenomena (Berkowitz, 2002;Dentz et al., 2011).
The fluid flow behavior, as a background information, is of importance in analyzing the conservative solute transport
behavior in rough fractures. To this end, the fracture roughness effect on the fluid flow has attracted much attention in the
last several decades (Zimmerman and Bodvarsson, 1996;Brush and Thomson, 2003;Brown, 1987). Based on the ideal
smooth parallel plate model, the classical cubic law is widely used to simplify fluid flow in single rough fractures.
However, many theoretical and experimental studies reported that the classical cubic law was valid when the flow was
linear (or the inertial effect was negligible) and may result in non-negligible errors when the fracture walls were rough.
To improve the performance of the cubic law, several studies (Zimmerman et al., 1991;Oron and Berkowitz, 1998)
noticed the influence of the local aperture and developed the well-known local cubic law that is able to explicitly consider



the spatial variability in aperture. Renshaw (1995) considered a correction factor based on fracture roughness to modify

the cubic law and showed that the modified cubic law was capable of describing a nonlinear relationship between the

fracture hydraulic and the mechanical apertures. Konzuk and Kueper (2004) evaluated various modifications of the cubic

law and found that even at the typical groundwater flow rate with the Reynolds number less than 1 ($Re < 1$), the flow rate

in single rough fractures was over-predicted at least 1.75 times by the local cubic law. A recent work reported by Wang et

al. (2015) showed that the accuracy of the local cubic law could be improved by consideration of the local tortuosity and

roughness. However, important characterizations of the nonlinear flow through rough fractures (i.e., eddies) was

neglected by Wang *et al*. (2015).

Several studies (Drazer et al., 2004;Jin et al., 2017;Kang et al., 2016) revealed that both flow and transport behaviors

are sensitive not only to roughness but also to the scale of roughness. According to the roughness definition by the

International Society of Rock Mechanics (ISRM) (Barton, 1978), the original fracture roughness can be also considered

as a two-scale combination (i.e., a small-scale unevenness superimposed on a large-scale waviness). The large-scale

waviness represents the primary waviness geometry (i.e., non-stationary primary roughness) and the small-scale

unevenness represents the secondary waviness geometry (i.e., stationary secondary roughness). With this conceptual

understanding, Zou et al. (2015) and Wang et al. (2016) investigated the importance of secondary roughness on the

nonlinear flow behavior in the two-dimensional (2D) and three-dimensional (3D) rough fractures, respectively. Since the

transport of a conservative solute is intrinsically coupled with fluid flow, our current interest is to further investigate



conservative solute transport behavior for two different scales of fracture roughness by using the roughness definition of
ISRM.
Fick's law has been widely used to describe solute transport for many decades (Fetter, 2000;Bear, 1972). The Fickian
transport model (i.e., advection-dispersion equation or ADE) always yields well-behaved breakthrough curves (BTCs).
However, a large body of evidence indicates that the anomalous (non-Fickian) BTCs associated with the early arrival and
long tails is ubiquitous in geological formations. This phenomenon is caused by the well-known non-Fickian transport
behavior, which is present even in a single fracture due to the fracture roughness (Thompson, 1991;Tsang and Tsang,
1989;Boutt et al., 2006). Cardenas et al. (2007) used the 2D Navier-Stokes (N-S) equation to simulate the solute transport
in a single rough fracture and showed that eddies observed at regions with relatively large apertures led to a power-law
tail in BTCs. Zheng et al. (2009) investigated the influence of aperture heterogeneity and anisotropy on solute transport
and showed that the dominant dispersion regime varied with statistical properties of aperture field such as mean, standard
deviation, and anisotropic ratio. Wang and Cardenas (2014) reported a detailed numerical investigation on the
relationship between the magnitude of fracture heterogeneity and non-Fickian transport behavior. Fiori and Becker (2015)
used a purely advective model to analyze the long tails of BTCs in a variable-aperture fracture based on the statistical
parameters in transmissivity field. Recently, Wang and Cardenas (2017) quantified the dependence of non-Fickian
transport on the fracture roughness with consideration of increasing fracture length scales. Without consideration of
roughness scales, Dou et al. (2018a) quantified the uniformity of concentration distribution based on the concept of



dilution index (Kitanidis, 1994) and their results showed roughness-induced eddies could provide resistance for the solute
transport.
To explain non-Fickian transport, several transport models were developed such as the continuous time random walk
(CTRW) (Berkowitz et al., 2006), the fractional advection-dispersion equation (FADE) (Benson, 1998), the mobile-
immobile model (MIM) (Van Genuchten and Wierenga, 1976), the equivalent-stratified medium (Nowamooz et al.,
2013), and the multi-rate mass transfer model (Haggerty, 2013). A detailed description and comparison of these transport
models can be found in several recent studies (Neuman and Tartakovsky, 2009;Gao et al., 2009). Fitting non-Fickian
BTCs to different transport models can accurately estimate the transport parameters and quantify the characterization of
the non-Fickian transport. The ADE has been inversely applied to estimate solute transport parameters in fractures for
many decades. However, the ADE model was proven to be incapable of explaining non-Fickian transport (i.e., scale-
dependent spreading, early arrival and long tails in BTCs) and consequently fitting non-Fickian BTCs using the ADE
model showed persistent errors (Becker and Shapiro, 2003;Bijeljic and Blunt, 2006;Briggs et al., 2014;Dou and Zhou,
2014;Jiménez‑Hornero et al., 2005).
Alternatively, some other non-Fickian models have been applied to fit the non-Fickian BTCs obtained in physical and
numerical experiments carried out in single fractures. Bauget and Fourar (2008) experimentally investigated non-Fickian
transport in a single rough fracture. Their results showed that the ADE model was unable to capture the long tail of BTCs
while the CTRW model was robust enough to model the non-Fickian transport due to the fracture roughness. Qian et al.



(2011b) studied the capacity of the MIM model for solute transport in a single fracture under non-Darcian flow
conditions. Their results showed that the MIM model captured both peak and tails in the BTCs better than the ADE
model. Recently, Cherubini et al. (2014) compared the performance and reliability of the MIM model and the explicit
network model (ENM) in a fracture network. Their results showed that although the ENM model better fitted BTCs
observed in a fracture network than the MIM model, the latter remained valid as well. This could be attributable to the
capability of the MIM model to intrinsically separate solute spreading into dispersion in the mobile region and mobile-
immobile mass transfer (Gao et al., 2009;Qian et al., 2011a).
The goal of this study is to investigate the influence of roughness on conservative solute transport in self-affine
fractures for two different scales of fracture roughness. Although Dou et al. (2018a) revealed that the roughness-induced
eddies, which could be considered as an immobile region in the MIM model, provided resistance for the solute transport,
the influence mechanism of the different scales of fracture roughness on the mass transfer between mobile and immobile
regions was not investigated. Moreover, our previous work (Dou et al., 2018b) based on CTRW model had demonstrated
the roughness scale dependence of the relationship between the longitudinal dispersion ($D_L$) and Peclet number ($Pe$), in
which the method of decomposing the fracture roughness into different scales was capable of providing an effective path.
Hence, we here focus on the differences in flow and solute transport at two different scales to provide a fundamental
understanding of the influence of roughness and physical insight into the inherent mechanism of transport, especially the
mass transfer between mobile and immobile regions, in single rough fractures. For this, we first generate self-affine
fracture walls by the successive random additions (SRA) procedure and then decompose the roughness of the synthetic



fracture into a primary roughness (large-scale) and a secondary roughness (small-scale) by a wavelet analysis technique.
The flow field and conservative solute transport in the original fracture and the primary fracture (without secondary
roughness) are simulated by coupling the N-S equation and ADE. Fluid flow is analyzed by Forchheimer's law and BTCs
are characterized by a proper inverse model (MIM model).

## 2. Construction and Decomposition of a Self-affine Fracture

### 2.1 Construction of a Self-affine Fracture

The surface of natural fractures typically follows a statistical self-affine distribution (Mandelbrot, 1983). Thus, to
construct a synthetic fracture, a self-affine fracture wall should be generated first. A 2D self-affine fracture wall height,
$Z(x)$, satisfies,

$$\lambda^H Z(x) = Z(\lambda x) \qquad (1)$$

where $H$ indicates the roughness magnitude or the so-called Hurst exponent, varying from 0 to 1 (dimensionless) and $\lambda$ is
a scaling factor (dimensionless). $Z(x)$ can be thought of as a function of an independent spatial or temporal variable $x$
(L). Here the successive random additions (SRA) method (Voss, 1988) is applied to generate self-affine walls. A more
detailed application for generating the self-affine wall by the SRA method can be found in our previous work (Dou et al.,
2013). It should be noted that to generate the self-affine wall by using the SRA method, the desired Hurst exponent must


be selected. Previous studies (Boffa et al., 1998;Schmittbuhl et al., 1993) on self-affinity of fractured rock showed that
the $H = 0.8$ could be as a characteristic Hurst exponent of self-affine fractures.

In this study, we use $H = 0.8$ as a characteristic Hurst exponent of self-affine fractures and used two different ways to

obtain the synthetic self-affine fracture with the generated self-affine walls. First, the generated self-affine wall $Z(x)$ is
considered as the lower fracture wall and the upper fracture wall is translated a vertical distance $\bar{b}$ without the relative
lateral displacement of $d_0$. As a result, the local aperture in the constructed self-affine fracture is constant and equal to $\bar{b}$
but the fracture wall is rough and self-affine (i.e. constant-aperture self-affine fracture), as shown in Figure 1 (a).
Alternatively, the two fracture walls can have a relative lateral displacement of $d_0$, which results in the fact that the two
fracture walls do not overlap (i.e. variable-aperture self-affine fracture), as shown in Figure 1 (b). Thus, the local aperture
$b(x)$ varies along the longitudinal direction, $b(x) = Z(x + d_0) - Z(x) + \bar{b}$. In this study, we consider a variable-aperture
self-affine fracture with $H$=0.8 and a total fracture length of 0.2 m as an example to illustrate the transport feature.

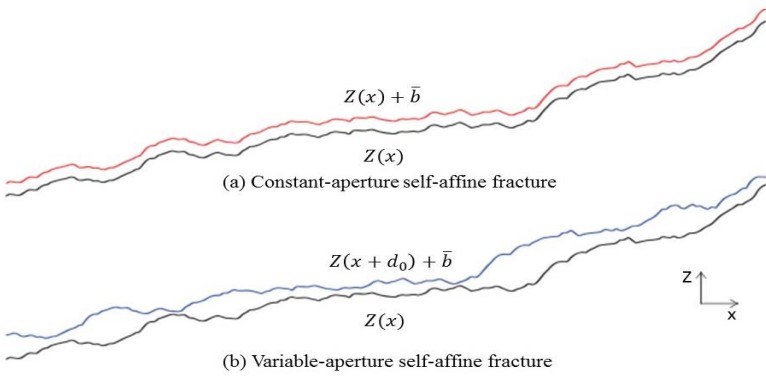






Figure 1. The construction process of synthetic self-affine fracture with $H$=0.8 and a total fracture length of 0.2 m.

**2.2 Decomposition of the Self-Affine Fracture**

The self-affine wall can be considered as signal profiles due to the local change of geometries at different scales. Here, $f(s)$ is defined as the fracture wall profile. Analogous to the signal profiles, the $f$ function representing the variation of the wall height is similar to the dependent variable in the frequency domain, while $s$ representing the variation of the distance along in-plane direction of fracture is similar to the dependent variable in the time domain. Previous studies (Wang et al., 2016;Zou et al., 2015) have shown that the wavelet analysis method is more suitable than other mathematical methods (i.e., image analysis and Fourier transform) for decomposing the self-affine roughness into primary (non-stationary) and secondary (stationary) roughness components. The non-stationary primary roughness represents larger scale undulations or waviness with the lower frequency, while the stationary secondary roughness represents the higher frequency variability. The wavelet transform of the self-affine fracture profile is given as,

$$W(a,\tilde{s}) = \int_{-\infty}^{+\infty} f(\tau)\psi_{a,s}(s)ds \quad (2)$$

where $\tilde{s}$ represents the horizontal distance, $a$ can be considered as a scaling parameter and represents the frequency of the height profile of the original wall, and $\psi_{a,s}(s)$ is the so-called mother wavelet function expressed as,

$$\psi_{a,s}(s) = a^{-\frac{1}{2}}\psi(\frac{s-\tilde{s}}{a}) \quad (3)$$



An appropriate mother wavelet function is important to decompose the roughness component of the self-affine fracture
profile.
In this study, we used the Daubechies Wavelet as the mother wavelet. The process of roughness decomposition is
realized by Mallat's pyramidal algorithm in a level-by-level procedure (Mallat, 1989). In each level, the self-affine
fracture wall is decomposed into low and high frequency components. The low frequency component is defined as an
approximate wall and represented the dominant geometrical formation of waviness or unevenness. The high frequency
component is defined as a detailed wall and represented the small-scale waviness or unevenness. As shown in Figure 2,
the decomposition of reconstruction of the original rough fracture walls are implemented at 8 levels, noted by A1-A8 and
D1-D8, respectively. For each level, the approximation walls, A$i$, represent the primary waviness (non-stationary primary
roughness) and the detailed walls, D$i$, represent the small-scale unevenness (stationary secondary roughness).
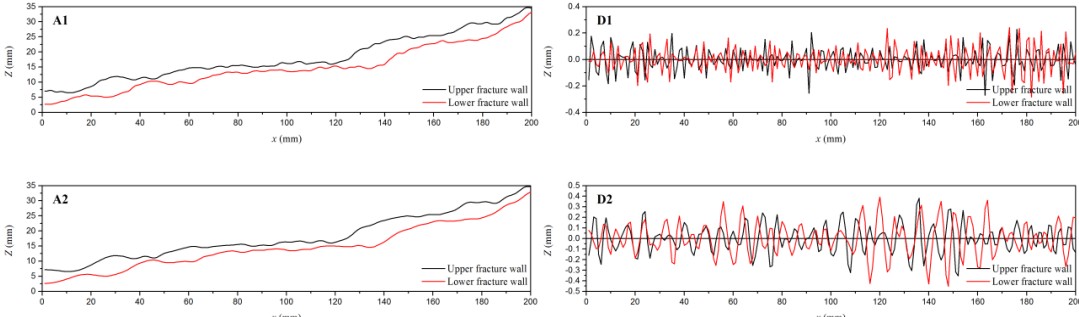













Figure 2. 8 levels decomposition of the roughness for the self-affine fracture with $H$=0.8. A1-A8 and D1-D8 represent the

approximate walls and the detailed walls, respectively.

During the process of roughness decomposition, it is crucial to determine an appropriate cut-off level for separating
roughness into the non-stationary primary roughness and the secondary roughness, where the non-stationary primary
roughness should be dominant for the geometric property, while the secondary roughness could be treated as a Gaussian
white noise process. Here, we use the quantitative criterion proposed by (Zou et al., 2015). In Zou's quantitative criterion,
the cut-off level is determined by analyzing the variance of the primary roughness, and locating where the variance of the
primary roughness changes significantly, while the secondary roughness could be considered as a Gaussian white noise
with a mean value close to zero.
Based on Zou's quantitative criterion, the cut-off levels for the fracture walls in Fig.2 are level 4 and level 5 for top
and bottom fracture walls, respectively. Consequently, the level 4 and level 5 approximation walls are considered as the
primary roughnesses for top and bottom fracture walls, respectively. The secondary roughnesses of top and bottom
fracture walls are determined by summing the level 1-4 and level 1-5 detailed walls, respectively. Hereafter, the
undecomposed fracture is denoted as the original fracture, while the decomposed fracture (with secondary roughness
removed) is denoted as the primary fracture.





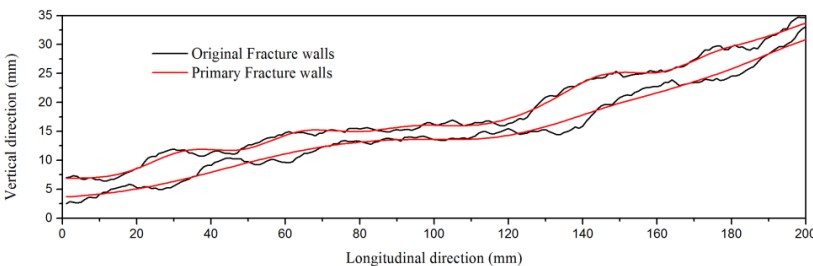

Figure 3. Comparison of original and primary fracture walls after the roughness decomposition. The primary fracture

walls only contain the non-stationary primary roughness of the original roughness where the stationary secondary

roughness is removed.

Table 1. The statistical properties of original and primary fractures

| Fracture | $\bar{b}$ (mm) | $\sigma$ (mm) | COV |
|---|---|---|---|
| Original | 3.3871 | 1.6502 | 0.4872 |
| Primary | 3.3701 | 0.9839 | 0.2920 |

Figure 3 shows the original fracture wall and the primary fracture wall after the roughness decomposition. It can be

seen that the primary fracture wall without the secondary roughness is smoother than the original fracture wall. The



statistical properties of the original and primary fractures are listed in Table 1. The decomposition of fracture roughness
has little influence on the average aperture but leads to a decreasing coefficient of variation (COV).

## 3. Direct Numerical Simulation: Fluid Flow and Conservative Solute Transport

### 3.1 Fluid flow

The fluid flow in single fractures is solved directly by the continuity equation (Eq. 4) and the N-S equation (Eq. 5)
under isothermal, incompressible, Newtonian, and steady-state flow conditions,

$$\nabla \cdot \boldsymbol{u} = 0 \quad (4)$$

$$-\nabla p = \rho(\boldsymbol{u} \cdot \nabla \boldsymbol{u}) - \nabla(\mu \nabla \boldsymbol{u}) \quad (5)$$

where $\boldsymbol{u} = [u, w]$ denotes the 2D velocity vector with components of $u$ and $w$, respectively ($LT^{-1}$), $\rho$ represents the
density of fluid ($ML^{-3}$), $\mu$ is defined as the dynamic viscosity of fluid ($ML^{-1}T^{-1}$), and $p$ is the fluid pressure
($ML^{-1}T^{-2}$).The standard water properties at 20℃ (i.e., $\rho$=998.2 kg/m$^3$ and $\mu$=1.002×10$^{-3}$ Pa·s) was used in this study.
The non-slip boundary conditions are applied for the fracture walls. As background flow, the steady-state flow is induced
by a given pressure gradient over the entire fracture. The COMSOL Multiphysics package (COMSOL Inc., Burlington,
MA, USA), a Galerkin finite-element software package, serves as a numerical solver for Eqs. (4) and (5). The calculating
domain for the fracture is discretized into 148,000 triangular elements (TE). A superfine mesh size (0.00015 mm) is
imposed near the fracture walls to capture the non-slip boundaries. The number of TE was determined by the mesh




independence analysis. Under the same pressure gradient (i.e., $-\nabla p$=200 Pa/m), the steady-state flow rate changes about
0.5% (from $1.215 \times 10^{-4}$ m$^3$/s to $1.221 \times 10^{-4}$ m$^3$/s) as the number of TE increases by about 100% (from 148,000 to
295,000), indicating that the 148,000 TE for the given fracture are sufficient to provide stable and accurate numerical
results. The dimensionless Reynolds number ($Re$) is given as ,
$$Re = \frac{\rho Q}{\mu W} \quad (6)$$

where $W$ denotes the width of fracture in the out of plane direction ($W$=1 m for the 2D problem) and $Q$ represents the
flow rate (L$^3$T$^{-1}$). We consider four different flow fields in both original and primary fractures where the $Re$ value varies
from 3.1 to 71.8.
**3.2 Conservative solute transport**

The conservative solute transport simulation is first attempted using the advection-diffusion equation,

$$\frac{\partial c}{\partial t} = -\nabla(\boldsymbol{u}c) + D_m \nabla^2 c \quad (7)$$

where $t$ represents time (T), $c$ denotes the conservative solute concentration (ML$^{-3}$), $D_m$ is defined as the diffusion
(L$^2$T$^{-1}$), and the 2D velocity tensor $\boldsymbol{u}$ is determined by the numerical solution of Eqs. (4) and (5). In this study, we
consider two different injection conditions for the solute transport simulation.



The first injection condition is simulated with the Dirac delta function. We assume that the concentrations of initial
instantaneous sources follow a sole function of the longitudinal coordinate $x_L^*$. Thus, the initial condition for Eq. (7) is
specified as,
$$c(\boldsymbol{x}, t = 0) = c_0(x_L^*) \quad (8)$$
where $c_0(x)$ is given by
$$c_0(x) = \begin{cases} \delta(t)\frac{m_0}{b(x)*\Delta L} & \text{if } x_L^* < x < x_L^* + \Delta L \\ 0 & \text{otherwise} \end{cases} \quad (9)$$
where $\delta(t)$ represents the Dirac delta function for the time variable, $m_0$ is the mass of injected solute (M), $b(x)$ is the
local aperture (L) and $\Delta L$ is the width of injected solute (L). $\Delta L$ is constant in all the simulations and $\Delta L/L = 0.001$,
where $L$ is the fracture length. The inlet and outlet boundary conditions are specified as,
$$c(0, t) = 0 \quad t \geq 0 \quad (10)$$
$$\partial c(L, t)/\partial n = 0 \quad t \geq 0 \quad (11)$$
where $n$ is the direction normal to the outlet boundary.
The second injection condition tested is continuous solute injection at the inlet boundary. This is the so-called step
injection condition. To do this, the inlet boundary condition in Eq. (10) becomes a Dirichlet boundary condition with the
assumption of step injection concentration with $c_0 = 1$ and is given by





$$c(0,t) = c_0 = 1 \quad t > 0 \quad (12)$$
where 1 in Eq. (12) represents a specified unit concentration.
The ratio of effluent solute mass to fluid mass is expressed as the BTCs,,
$$c_f = \frac{\int_0^b ucdz}{\int_0^b udz} \quad (13)$$
and the normalized BTCs and time are respectively,
$$c' = c_f/c_0 \quad (14)$$
$$t' = Qt/AL \quad (15)$$
Where $A$ is the area of the 2D fracture ($L^2$), $t'$ is the dimensionless time or number of pore volume (PV), and $c'$ is the
dimensionless concentration. The mesh used in solute transport simulation is the same as the fluid flow simulation. The
numerical dispersion can be negligible depending on sensitivity analysis of mesh size and time step.
In this study, we define the Peclet number ($Pe$), $Pe = \tau_D/\tau_u = \bar{u}\bar{b}/D_m$, as the ratio of the characteristic diffusive
time ($\tau_D = \bar{b}^2/D_m$) to the characteristic advective time ($\tau_u = \bar{b}/\bar{u}$). To investigate the independent influence of
multiscale roughness, rather than the combined influence of multiscale roughness and fracture length scale on the
transport, we consider the case of solute transport at the potential asymptotic time scale. In such a condition, anomalous
early arrival and long tails in BTCs, which is the classical non-Fickian behavior, is caused by the influence of roughness
rather than the pre-asymptotic transport of a conservative solute due to the limitation of the fracture length scale. Based
on the equivalent 2D ideal model (Wang et al., 2012), we calculate the potential asymptotic time (T) for the self-affine
fractures. Following Taylor's seminal work (Aris, 1956), the potential critical time (T) is given as



$$T_t = \bar{b}^2/4D_m \quad (16)$$

At times larger than $T_t$, the transverse diffusion gradient eventually leads to complete cross-sectional homogenization of
the solute distribution. For a given fracture, the critical mean advective velocity ($LS^{-1}$) can be obtained by
$$U_a = L/T_t = 4D_m L/\bar{b}^2 \quad (17)$$

whereas the corresponding critical advective time (T) is
$$T_a = \bar{b}/U_a = \bar{b}^3/4D_m L \quad (18)$$

Depending on the definition of $Pe$, the Peclet number corresponding to the critical advective time, denoted as $Pe_a$, for
asymptotic solute transport is obtained as,
$$Pe_a = \tau_d/T_a = 4L/\bar{b} \quad (19)$$

It can be seen from Eq.(19) that $Pe_a$ is only dependent on the geometry of fracture. In this study, $Pe_a$ is equal to 237
and the solute transport therefore is simulated under the conditions of $Pe$ =8.8, 81.7, 147.6, and 205.8, respectively. This
implies that all of simulated solute transport is asymptotic and the non-Fickian transport behavior could be considered as
a result of the influence of roughness rather than the length of fracture.
**4. Inverse Model for Solute Transport**
**4.1 Fickian transport model**



Based on Fick's law, the advection-diffusion equation (Eq.(7)) can be transformed to ADE by replacing the velocity
vector $\boldsymbol{u}$ by the average velocity $\bar{u}$. For the step injection condition, the corresponding analytical solution for the ADE
model is given by

$$C(x,t) = \frac{C_0}{2}\left[erfc\left(\frac{x-\bar{u}t}{2\sqrt{D_f t}}\right) + \exp\left(\frac{\bar{u}x}{D_f}\right)erfc\left(\frac{x+\bar{u}t}{2\sqrt{D_f t}}\right)\right] \quad (20)$$

where $D_f$ is the fitted dispersion coefficient ($L^2 T^{-1}$). For the Dirac delta injection condition, the corresponding analytical
solution for the ADE model is given by

$$c(x,t) = \frac{m_0}{A}\frac{x}{\sqrt{4\pi D_f t}}exp(-\frac{(x-\bar{u}t)^2}{4D_f t}) \quad (21)$$

**4.2 MIM model**
Unlike the Fickian model, the single-rate MIM model describe the Fickian transport in the mobile domain and
diffusion-driven mass exchange between the mobile and immobile domains. For the conservative solute transport, the
MIM model is given as,

$$\theta_m\frac{\partial c_m}{\partial t} = \theta_m D_{f,m}\frac{\partial^2 c_m}{\partial x^2} - \bar{u}\theta_m\frac{\partial c_m}{\partial x} - \alpha(c_m - c_{im}) \quad (22)$$

$$\theta_{im}\frac{\partial c_{im}}{\partial t} = \alpha(c_m - c_{im}) \quad (23)$$



where $c_m$ and $c_{im}$ are defined as the solute concentrations in the mobile and immobile domains, respectively; $\theta_m$ and $\theta_{im}$
represent the water contents in the mobile and immobile domains, respectively, and the sum of $\theta_m$ and $\theta_{im}$ is equal to
one for a saturated 2D fracture; $D_{f,m}$ represents the dispersion coefficient and $\alpha$ is the mass transfer rate between the
mobile and immobile domains ($T^{-1}$). A dimensionless $\beta = \theta_m/(\theta_{im} + \theta_m)$ is introduced as the mobile water fraction. It
should be mentioned that our motivation to implement the MIM model is based on the assumption that the potential eddy-
controlled domain can be considered as an immobile domain, rather than considering the fracture matrix as an immobile
domain. Thus, we assume that the fracture matrix is impermeable.
**4.3 Inverse parameter estimation from BTC**
BTCs from numerical simulations can be used to estimate parameters used in the ADE or MIM model. The goodness-
of-fitting was quantified by introducing a global error, $E_i$, which is given as,
$$E_i = \sqrt{\sum_i^N (c_{fit,i} - c_{model})^2/N} \quad (24)$$

where $c_{fit,i}$ ($i$=1, 2, 3, , , $N$) represents concentration in the fitted BTCs, and $c_{model}$ is the BTCs from the numerical
simulations, $i$ refers to either the ADE or MIM model, and $N$ represents the number of calculated data in BTCs. Global
errors from the ADE and MIM models hereafter are denoted as $E_{ADE}$ and $E_{MIM}$, respectively. Both of the velocity and
dispersion coefficient are considered as the fitted parameters in the inverse parameter estimation with the ADE model.
There are four initial input parameters, $\beta$, $\alpha$, $\bar{u}_{f,MIM}$, and $D_{f,MIM}$ in the inverse parameters estimation with the MIM



model. Accurate initial guesses for fitted parameters would be beneficial to improve the convergence of the inverse model
(Wang and Cardenas, 2014;Wang and Cardenas, 2015). The initial values of velocity $\bar{u}_{f,MIM}$ and dispersion coefficient
$D_{f,MIM}$ are set equal to the counterparts from the ADE model. The 1STOPT code and STANMOD V2.08 based on
CXTFIT code (Toride et al., 1995) are employed for the inversion of the ADE and MIM models, respectively.

## 5. Results and Discussion

### 5.1 Flow fields in original and primary fractures

Flow at four different Reynolds numbers (*Re*=3.1, 28.5, 51.5, and 71.8) is simulated in the original and primary
fractures. The flow fields in the original fracture and the primary fracture are denoted as OF#1-OF#4 and PF#1-PF#4,
respectively. Without loss of generality, Figure 4 only illustrates the numerical results of OF#1, OF#4, PF#1, and PF#4
where the influence of Reynolds numbers and roughness on the flow fields can be observed. Comparing Figs.4 (a) and 4
(c), the flow velocity distribution is more heterogeneous in the original fracture than in the primary fracture. For both of
the original and primary fractures, the relatively high local flow velocity is captured at small-aperture segments, while the
relative low local flow velocity is observed at the abruptly changing large-aperture segments where the eddies form as the
Reynolds number increases. The streamlines are smoother in the primary fracture than in the original fracture, suggesting
that the tortuosity of the flow path is significantly influenced by the secondary roughness. As the Reynolds number
increases, eddies form and develop at the abruptly changing large-aperture segments. As a result, the effective advection
channel becomes narrow. Several previous works (Bouquain et al., 2012;Briggs et al., 2016;Qian et al., 2012) also





showed the similar results that increasing Reynolds number causes the growth of eddies. However, comparing Figs.4 (b)
and 4 (d) for the primary fracture where the secondary roughness is not included, there are no visible eddies and the
streamlines have the same patterns with cases at low Reynolds number (Fig. 4 (c)). This indicates that the secondary
roughness is primarily responsible for the presence of eddies.

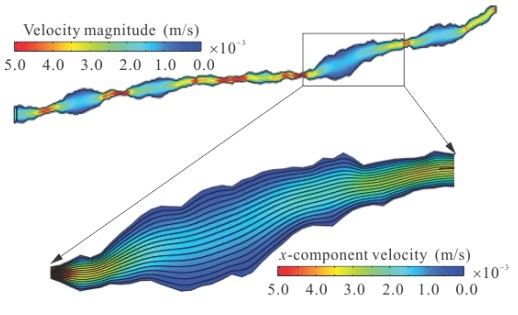

(a) Original Fracture with $Re$=3.1

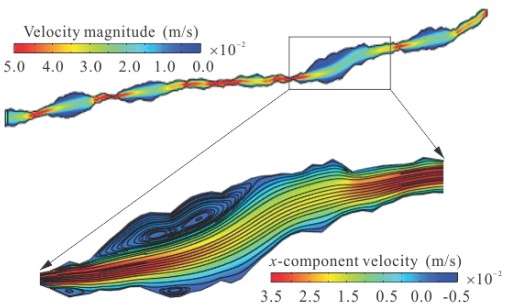

(b) Original Fracture with $Re$=71.8

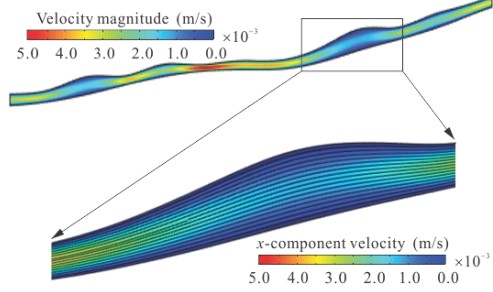

(c) Primary Fracture with $Re$=3.1

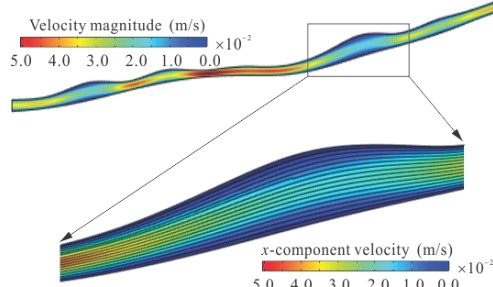

(d) Primary Fracture with $Re$=71.8






Figure 4. The non-uniform flow field in the self-affine fracture under the different Reynolds number. (a)-(b) and (c)-(d)
represent the flow fields in the original and primary fractures with *Re*=3.1 and 71.8, respectively. The entire view of the
fractures shows the velocity magnitude, while the corresponding enlarged segment shows the streamlines and *x*-
component velocity distribution.

It should be mentioned that the presence of eddies cannot be used to determine if the flow regime is non-Darcian

(nonlinear flow) or Darcian (linear flow). Since the discrepancies of solute transport behavior are captured under different
flow regimes (Lee et al., 2015;Qian et al., 2011b;Qian et al., 2011a), it is necessary to determine when the flow regime is
linear or nonlinear. For determining the flow regime in the simulations, Forchheimer's law (Jacob, 1972) is applied to
determine the critical Reynolds number ($Re_c$) that is proposed for characterizing the onset of flow transition to
nonlinearity (Javadi et al., 2014). If the Reynolds number of fluid flow is less than $Re_c$, the fluid flow can be considered
as linear. Otherwise, the fluid flow is the nonlinear. Forchheimer's law is:

$$-\Delta p = aQ + bQ^2 \quad (25)$$

where $a = 12\mu/We_h^3$ ($MT^{-1}L^{-5}$) and $b = \beta\mu/W^2 e_h^2$ ($ML^{-8}$) denote the linear and nonlinear coefficients, respectively,
and $\beta$ and $e_h$ represent the Forchheimer coefficient and hydraulic aperture, respectively. When the inertial effect of fluid
flow becomes negligible at low velocity, the nonlinear term (the second term on the right side of Eq. (25)) tends to zero
and Forchheimer's law reduces to Darcy's law for linear flow. $Re_c$ is given by

$$Re_c = \frac{a\rho\alpha}{b\mu W(1-\alpha)} \quad (26)$$



where $\alpha$ is a nonlinear effect factor denoted as the ratio of the nonlinear pressure drop to the overall pressure drop, $\alpha =$
$bQ^2/(aQ + bQ^2)$. In this study, a critical value of $\alpha =$5% is used to quantify the transition of flow regimes from linear to
nonlinear. This critical value for $\alpha$ is consistent with previous studies (Wang et al., 2016;Zhou et al., 2015).
Table 2. Critical $Re$ and best-fitted results from Forchheimer equation.

| Fracture | $a$ (kg·s$^{-1}$·m$^{-5}$) | $b$ (kg·m$^{-8}$) | $Re_c$ |
|---|---|---|---|
| Original | $1.59\times10^6$ | $8.30\times10^9$ | 10.1 |
| Primary | $4.12\times10^5$ | $5.22\times10^8$ | 41.4 |


Table 2 lists the critical Reynolds number and the best-fitted results for the parameters ($a$ and $b$) in the Forchheimer
equation for the original and primary fractures. The critical Reynolds number is smaller in the original fracture than in the
primary fracture, which indicates that the secondary roughness significantly impacts the flow behavior. Thus, OF#1
($Re$=3.1), PF#1($Re$=3.1), and PF#2($Re$=28.5) are in the linear flow regime, while the rest (OF#2-OF#4 and PF#3-PF#4)
are in the non-linear flow regime.
**5.2 BTCs and RTDs**
To analyze the influence of secondary roughness on solute transport, the flux-weighted BTCs and RTDs are
calculated from numerical simulations involving different Peclet numbers. Figure 5 shows the calculated BTCs in the
original and primary fractures. In general, the apparent differences in BTCs between the original fracture and the primary
fracture indicates that the secondary roughness plays a crucial role in the solute transport. The early arrival and long tail




as non-Fickian characteristics can be observed in all of the BTCs, indicating that the transport is non-Fickian. It should be
noted that all of the Peclet numbers used in this study are less than the critical Peclet number for asymptotic solute
transport. Thus, the fracture roughness rather than the length of the fracture leads to the non-Fickian transport behaviors.

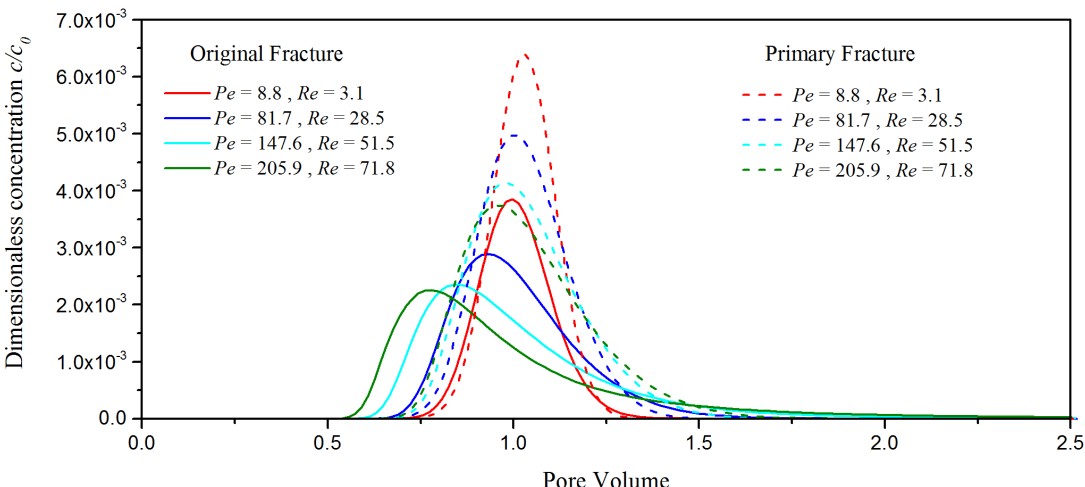


Figure 5. The breakthrough curves under the different Peclet numbers at the outlets of the original fracture and the
primary fracture.

In the original fracture (see Figure 5), the early arrival and long tail in the BTCs are enhanced as the Peclet number

increases. Similar trends can be found in the primary fracture where the secondary roughness is excluded. However, the
non-Fickian characteristics of BTCs are stronger in the original fracture than in the primary fracture, suggesting that the
secondary roughness significantly enhances the non-Fickian characteristics of BTCs. This can be attributed to the fact



that the secondary roughness leads to enhanced nonlinear flow behavior. Another interesting finding is the peak
concentration of BTCs. As the Peclet number and Reynolds number increase, the peak concentrations of the BTCs
decrease. This trend is observed in both the original and primary fractures. This could be qualitatively explained from the
view of the solute dilution (mixing) process in the sense that greater dilution leads to a lower peak concentration.
Comparing BTCs in the original and primary fractures, the peak concentrations of BTCs are lower in the original fracture
than in the primary fracture for the case with the same Peclet number. This implies that the solute dilution process is
significantly enhanced by the secondary roughness. Although beyond the scope of the current study, it is worth noting
that the magnitude of the solute dilution could be analyzed quantitatively using the concept of a scalar dissipation rate
(Dreuzy et al., 2012;Le Borgne et al., 2010) and a statistical-entropy dilution index (Kitanidis, 1994).





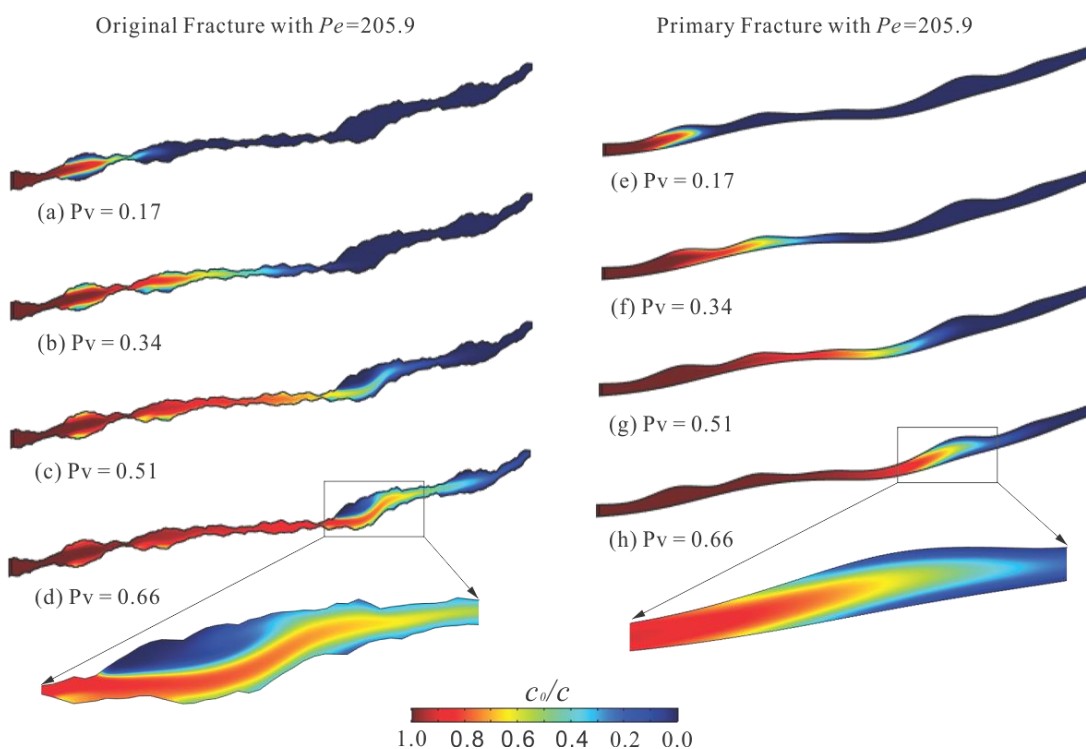

Figure 6. The spatial evolution of solute concentration distribution under Pe=205.9 in the original fracture ((a)-(d)) and

the primary fracture ((e)-(h)), respectively.

Figure 6 shows the spatial evolution of solute concentration distribution for $Pe$=205.9 in the original and primary

fractures. This evolution is found to be sensitive to the aperture. The solute is transported slower near the fracture walls

than in the middle clearly showing that there is an effective advection channel along the longitudinal direction for solute

transport. Comparing Figures 6 (a) and (b), the secondary roughness has an important influence on spatial evolution of





solute concentration distribution. The effective advection channel is narrower and more tortuous in the original fracture
than in the primary fracture due to the fact that the growing eddy flow region is mainly due to the secondary roughness.

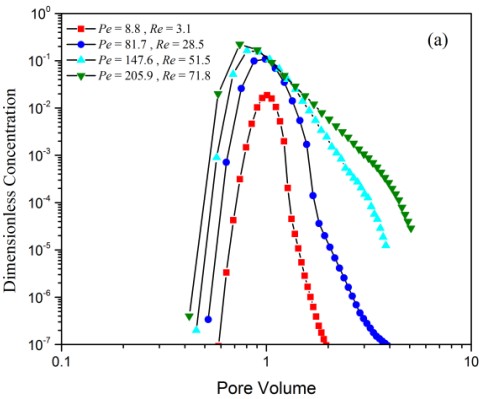 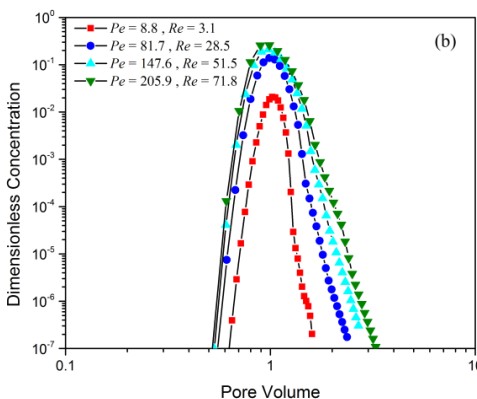


Figure 7. The RTDs under different Peclet numbers at the outlets of the original fracture (a) and the primary fracture (b),
respectively.

RTDs were calculated for solute transport in both the original and primary fractures for characterizing the long tails of

BTCs. Figs. 7 (a) and (b) show RTDs for different Peclet numbers in the original fracture and the primary fracture,
respectively. The tails after the peak are apparent in the RTDs and follow a power-law drop (especially for the large $Pe$).
The degree of power-law drop increases as the Peclet number increases. Comparing RTDs in the original and primary
fractures, removing the secondary roughness from the original fracture roughness leads to shortening tails. This could be
attributed to the secondary roughness enhancing the nonlinear flow behavior.


**5.3 Inverse MIM and ADE models**

To further investigate the fracture multiscale roughness influence on the transport behavior, the ADE and MIM model are used to fit all of the BTCs in the flow fields OF#1-OF#4 and PF#1-PF#4, respectively. Figure 8 shows the best-fitted results of BTCs using the ADE and MIM in the OF#1-OF#4. The ADE model is incapable of capturing the peak concentration and tails of BTCs. In contrast, the best-fitted results of BTCs from the MIM model do not exhibit a Gaussian-shaped distribution, showing that the MIM model is more capable than the ADE model for describing the roughness-induced non-Fickian characteristics of BTCs, especially the peak concentration and tail. This is also reflected by the coefficient of determination $R^2$ and the global error ($E_i$), as shown in Table 3.

Although the best-fitted results of BTCs are better from the MIM model than from the ADE model (Table 3), the results from both the ADE and MIM models tend to yield an increasing global error as the Peclet number and Reynolds number increase. This is reasonable because as the Reynolds number increases, the spreading and mixing of the solute would be significantly influenced by the enhanced non-linear flow behavior (i.e., eddies). In such a condition, the MIM model considering a single-rate exchange process between the immobile and mobile domains may be inadequate (Dentz and Berkowitz, 2003). Figures 8c and 8d show the deviations of MIM model to direct simulation results. As the Peclet number increases, the enhanced non-linear flow behavior could induce a distribution of eddies of evolving sizes. As a limitation of the single-rate exchange process in the MIM model, this evolution of the sizes of eddies with Peclet number for the same fracture may not be reflected by the MIM model, which has also been found in porous media (Dreuzy and Carrera, 2016). Our primary aim is not to analyze the validity of transport model but to emphasize the influence of the





fracture multiscale roughness on the transport mechanism. Comparing the results in Figures 8 and 9, the best-fitted results
of the ADE and MIM models are better for the primary fracture than for the original fracture, which reflects the impact of
secondary roughness on non-Fickian behavior and deviation from the ADE and MIM models.

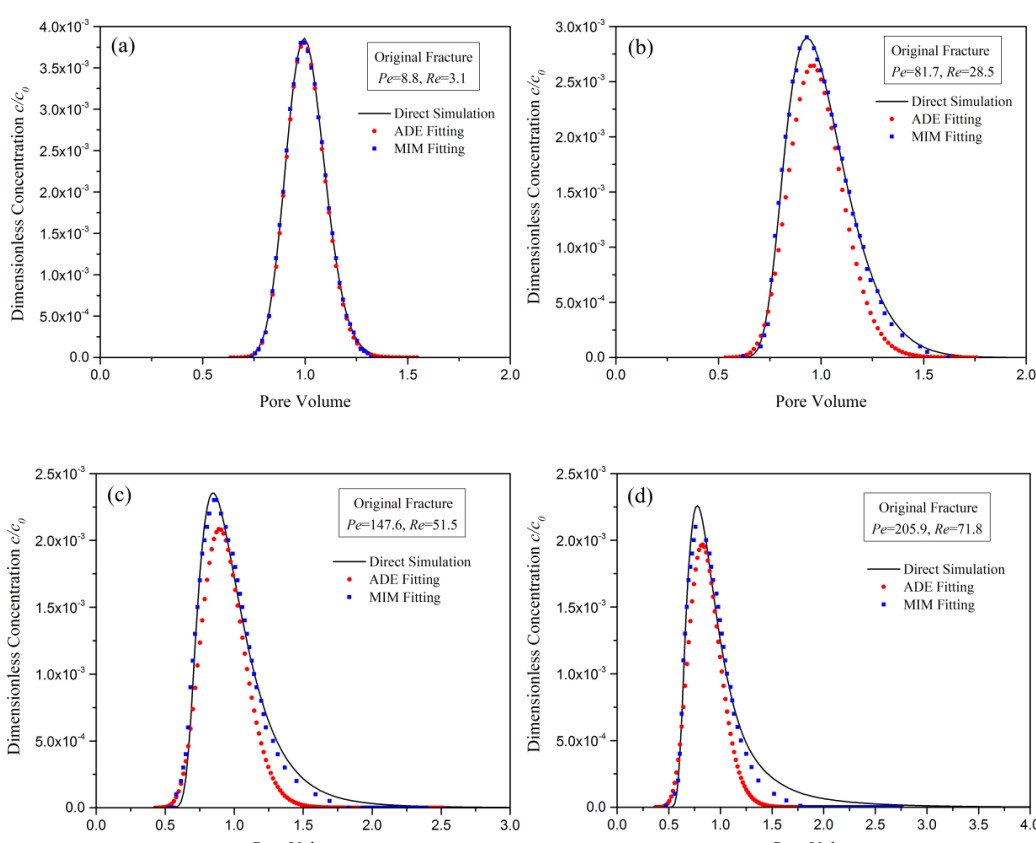





Figure 8. Best-fitted results of BTCs at different Peclet number and Reynolds number using the ADE and MIM models in
the original fracture.

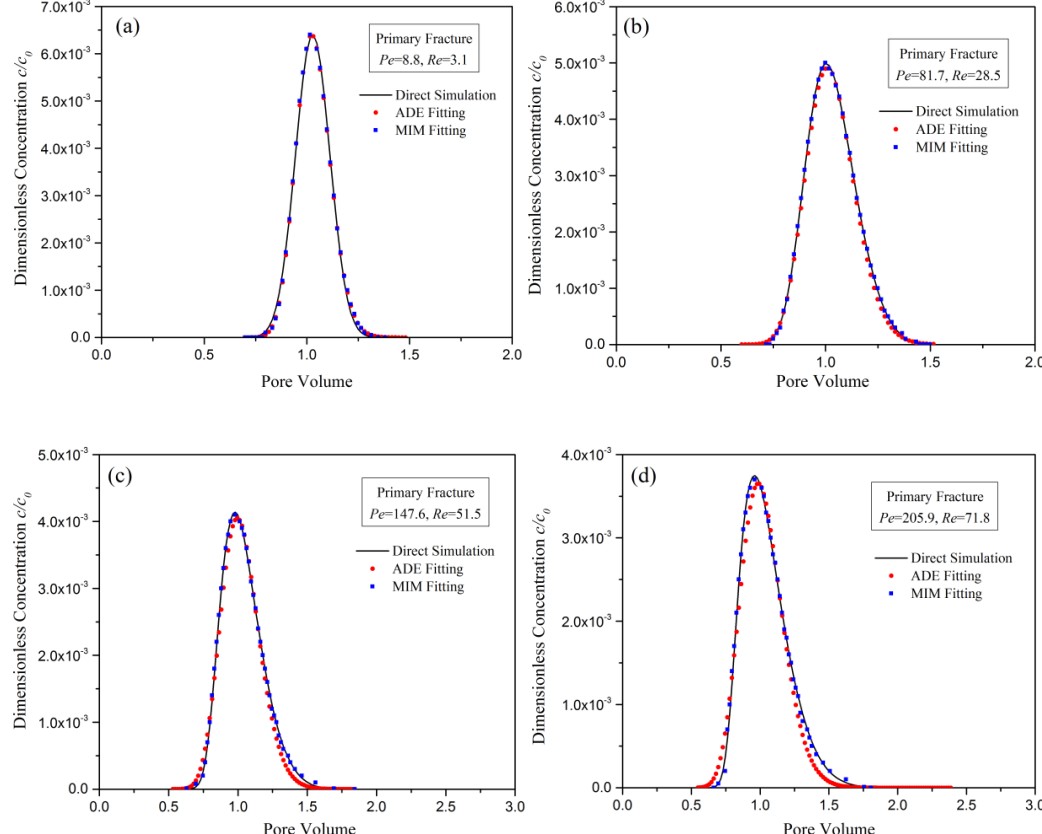

Figure 9. Best-fitted results of BTCs at different Peclet number and Reynolds number using the ADE and MIM models in
the primary fracture.





Table 3. Statistical results from the goodness-of-fitting of the ADE and MIM models.

| Flow Field no. | $Pe$ | $Re$ | MIM model | | ADE model | |
|---|---|---|---|---|---|---|
| | | | $R^2_{MIM}$ | $E_{MIM}$ | $R^2_{ADE}$ | $E_{ADE}$ |
| OF#1 | 8.8 | 3.1 | 0.9999 | 0.0003 | 0.9992 | 0.000304 |
| OF#2 | 81.7 | 28.5 | 0.9992 | 0.0004 | 0.9994 | 0.002 |
| OF#3 | 147.6 | 51.5 | 0.9916 | 0.000927 | 0.9872 | 0.002181 |
| OF#4 | 205.9 | 71.8 | 0.9804 | 0.001285 | 0.9771 | 0.002371 |
| PF#1 | 8.8 | 3.1 | 0.9983 | 0.000707 | 0.9992 | 0.000679 |
| PF#2 | 81.7 | 28.5 | 0.9989 | 0.000224 | 0.9994 | 0.000722 |
| PF#3 | 147.6 | 51.5 | 0.9975 | 0.000361 | 0.9972 | 0.001161 |
| PF#4 | 205.9 | 71.8 | 0.9905 | 0.000529 | 0.9945 | 0.001491 |

Table 4. Estimated values of parameters for the MIM and ADE models at different Peclet numbers and Reynolds

numbers in both original and primary fractures.

| Flow Field no. | $Pe$ | $Re$ | MIM model | | | | ADE model | |
|---|---|---|---|---|---|---|---|---|
| | | | $\beta$ | $\alpha^{*a}$ | $D_{f,MIM}/D_{Taylor}$, $(\%_E)^b$ | $\bar{u}_{f,MIM}/\bar{u}^c$ | $D_{f,ADE}/D_{Taylor}$, $(\%_E)^b$ | $\bar{u}_{f,ADE}/\bar{u}^c$ |
| OF#1 | 8.8 | 3.1 | 0.88 | 3.12 | 1.26 (26%) | 1.03 | 1.38 (38%) | 1.08 |
| OF#2 | 81.7 | 28.5 | 0.71 | 4.25 | 1.17 (17%) | 1.05 | 1.28 (28%) | 1.11 |
| OF#3 | 147.6 | 51.5 | 0.67 | 3.14 | 0.92 (8%) | 1.09 | 1.12 (12%) | 1.16 |
| OF#4 | 205.9 | 71.8 | 0.61 | 5.72 | 0.81 (19%) | 1.12 | 0.95 (5%) | 1.28 |
| PF#1 | 8.8 | 3.1 | 0.94 | 5.84 | 1.11 (11%) | 1.02 | 1.25 (25%) | 1.04 |
| PF#2 | 81.7 | 28.5 | 0.86 | 6.02 | 1.01 (1%) | 1.02 | 1.17 (17%) | 1.09 |
| PF#3 | 147.6 | 51.5 | 0.78 | 4.85 | 0.98 (2%) | 1.03 | 0.99 (1%) | 1.11 |
| PF#4 | 205.9 | 71.8 | 0.72 | 6.51 | 0.92 (8%) | 1.05 | 1.02 (2%) | 1.14 |

[a] $\alpha^*$ denotes the dimensionless mass transfer coefficient normalized by the characteristic advection time scale, $\alpha^* =$
$\alpha\tau_D = \alpha\bar{b}/\bar{u}$.
[b] $\%_E$ denotes the percent error between the fitted dispersion and Taylor dispersion.





$^{c}$ $\bar{u}$ denotes the mean velocity calculated from the direct simulations.

The best-fitted velocities of the ADE and MIM models are larger than the mean flow velocity calculated from the

numerical simulations (Table 4). This is mainly due to the fact the transport velocity along the middle of the fractures is
faster than the mean flow velocity. This deviation of the fitted velocity from the mean flow velocity increases as the
Peclet number and Reynolds number increase, since the nonlinear flow behavior (i.e., eddies) increases the variability of
local flow velocity. As expected, the fits of ADE transport velocity to numerical results are better in the primary fracture
than in the original fracture at the same Peclet number and Reynolds number. This indicates that the secondary roughness
enhances the channelling impact on solute transport.

For the asymptotic solute transport, the effective dispersion coefficient can be directly predicted by the Taylor

dispersion. For transport through an ideal single fracture with smooth parallel walls, the fluid flow is stratified
(Posieuille), and the Taylor dispersion is defined as,

$$D_{Taylor} = \frac{(\bar{u}\bar{b})^2}{210 D_m} + D_m = \frac{Q^2}{210 D_m} + D_m \quad (27)$$

Depending on Eq.(27), we can directly estimate $D_{Taylor}$ using $Q$ and $D_m$ without fitting the BTCs. It should be noted that
the Eq.(27) is only valid for describing solute transport after the asymptotic time and length scale, at which the spreading
of the solute reaches its asymptotic status. This requires that the Peclet number should be less than the critical Peclet
number (Eq. (19)) for a given 2D fracture. Since the Peclet number used in this study is less than the critical Peclet
number, one can expect that the fitted effective dispersion would be equal to the Taylor dispersion in the same fracture





with the same Peclet number. Therefore, comparing terms $D_{f,i}$ and $D_{Taylor}$ in Table 4 with the same Peclet number in
different fractures can be interpreted as a metric for the influence of roughness on the solute dispersion.
Although previous studies (Detwiler et al., 2000;Roux et al., 1998) have shown that the Taylor dispersion was valid
under relatively high Peclet numbers, our results here show that the values of $D_{f,i}/D_{Taylor}$ are fluctuating around unity,
indicating that the Taylor dispersion either underestimates or overestimates the fitted dispersion from the ADE and MIM
models. This is consistent with previous studies (Wang and Cardenas, 2014), where such a fluctuating difference is
considered as a result of the non-Fickian transport and the influence of fracture heterogeneity and flow inertia forces.
Another observation from Table 4 is that the percent errors between the fitted dispersion and the Taylor dispersion are
much smaller in the primary fracture than in the original fracture. This further indicates that the secondary roughness is a
primary factor that causes the non-Fickian transport.
Analyzing the estimated parameters $\beta$ and $\alpha^*$ from the MIM model in both the original and primary fractures can
provide insight into the mechanism of the impact of secondary roughness enhancement on non-Fickian transport. The
estimated parameter $\beta$ decreases while the $\alpha^*$ generally increases as the Peclet number and Reynolds number increase in
the original and primary fractures, indicating that the immobile domain fraction increases and the mass exchange process
between immobile and mobile domains is enhanced.
It is also clearly shown in Figs.4 and 6 that eddies resulting from the nonlinear flow act as an immobile domain and
the mixing and spreading processes of the solute are significantly delayed. An increasing flow velocity results in a higher



contact area between the immobile and mobile domains, which in turn gives a rise to the enhanced solute mixing between
these two domains (Cherubini et al., 2014;Gao et al., 2009).  As the secondary roughness is removed, the influence of
nonlinear flow characteristics (i.e., eddies and streamline tortuosity) on solute transport is significantly reduced. As a
result, the $\beta$ value is smaller while the $\alpha^*$ value is larger in the primary fracture than in the original fracture at the same
Peclet number and Reynolds number.

## 6. Summary and Conclusion

To investigate the influence of multiscale roughness on the flow and conservative solute transport in the self-affine
fractures, we constructed the self-affine fracture by the SRA and decomposed the original fracture roughness into large-
scale (primary roughness) and small-scale (secondary roughness) roughness. The numerical simulations of flow and
solute transport were conducted in the original fracture and primary fracture (without the secondary roughness) with a
varying range of Peclet number and Reynolds number. This enabled us not only to analyze the flow and solute transport
with consideration of Peclet number and Reynolds number, but also to emphasize the influence of multiscale roughness.
The fluid flow was characterized by Forchheimer's law and the results showed that the secondary roughness
significantly enhanced the nonlinear flow (i.e., eddies and tortuous streamlines). Since the simulations of the solute
transport was controlled under the asymptotic transport regime in which the simulated $Pe$ was less than the critical $Pe$,
any Non-Fickian transport could be considered as a result of fracture roughness rather than the fracture length. BTCs and
RTDs showed that Non-Fickian (i.e., early arrival and long tails) became weak as the Peclet number and Reynolds



number increased in both the original and primary fractures. However, it was found that the Non-Fickian behavior was
stronger in the original fracture than in the primary fracture where the secondary roughness was excluded. This indicated
that the secondary roughness significantly enhanced the Non-Fickian transport. A qualitative analysis showed that the
peak concentration in BTCs decreased as the secondary roughness was removed. This implied that the secondary
roughness could be curial for enhancing the solute dilution.
It was confirmed through model fitting that the ADE model was incapable of describing the roughness-induced non-
Fickian transport even at the asymptotic time, while the MIM model exhibited better results than the ADE model. The
estimated parameters from these two models showed that due to the influence of channelling flow, the fitted mean
velocity was larger than the mean flow velocity from the numerical simulations. After removing the secondary roughness,
we found that the deviation of the fitted velocity from the mean flow velocity was decreased. The mobile domain fraction
decreased while the mass exchange rate generally increased between the immobile and mobile domains as the Peclet
number and Reynolds number increased. The secondary roughness resulted in the decreasing mobile domain fraction and
the increasing mass exchange rate.
Our study provides insight into the influence of multiscale roughness on flow and solute transport behavior in
fractures. The current works and the idea of decomposing the fracture roughness into two different scales may also be
important to upscale roughness-induced non-Fickian transport. Further investigation is needed to extend this study to
conservative and reactive solute transport with the consideration of three-dimensional fractures and the pre-asymptotic
time scale.





## Acknowledgement

The work is supported by the National Natural Science Foundation of China (Nos.41602239 and 41572209), the

Fundamental Research Funds of the Central Universities (No.2016B05514), the International Postdoctoral Exchange

Fellowship Program (No.20150048) and the Natural Science Foundation of Jiangsu Province (No.SBK20160861).

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
