# Peer review of "Multiscale Roughness Influence on Conservative Solute Transport in Self-affine"

_Hydrology and Earth System Sciences, 2018_

## Referee Comment (RC1) · Anonymous Referee #1 · 7 Jul 2018

This paper simulates flow and transport in a two dimensional self affine fracture and then explores the use of a mobile-immobile model (MIM) to match observed transport behavior as characterized by breakthrough curves.

Overall the work in the paper appears to be correct, although I have some minor concerns as listed below, I do not believe that this paper rises to the standard that I would expect from a publication in a journal like HESS. HESS (and WRR) are the two highest impact journals in hydrology and as such I hold papers that are submitted there to a higher overall standard. The work in this paper is mostly incremental in nature and it is not clear that there is much real added value science here, given that we know many of the things that the authors point out – it is no surprise that transport in a complex fracture is not exactly well described by Taylor dispersion (for a parabolic flow) and that

some anomalous behaviors associated with the more complex makeup will emerge. However, I believe that we are well beyond the years of where we just fit anomalous transport model parameters to data and show better fits. Unless you can explicitly tie the anomalous transport parameters to physical and quantifiable characteristics of the fracture and/or flow I am not sure that there is enough value added. If you could, then that would be a sufficient advance in my view to warrant publication. Specific Comments

- From other papers I have seen this research group has used this same setup to study a variety of flow and transport problems. While I have absolutely no problem with this I do think it is important that it all be put in the context of state of the art models and such a 2d model falls somewhat short there – e.g. Peter Kang (formerly MIT now Minnesota) has several papers looking at anomalous transport in realistic three dimensional fractures. This is just one example of many others that are not touched on in this work. - Throughout the paper the quality of writing and English is poor and needs substantial revision. While it is possible to read the paper and understand everything the standard falls short and is a problem. - On page 17, if the continuous simulations yield results that are not simply the integratred convolution of the pulse additions, something fishy must be going on since this system is perfectly linear. Why look at both conditions? - Line 254 – just stating numerical diffusion is not a problem is not convincing to me. The numerical solutions from COMSOL are known to have potentially large numerical diffusion and so it would be nice to have some 'proof' of this statement. - I find the entire discussion on Forcheimer flow to be irrelevant to the central message of this work here. Also, eddies are not just a signature of nonlinear effects (see for example the works of Dykaar & Kitanidis (WRR 1996, TIPM 1997) who show their emergence in Stokes flow). Unless this can explicitly be linked to the anomalous transport parameters of the MIM it seems very tangential. - The Taylor dispersion discussion is also not surprising. Boundary fluctuations are known to change dispersion effects – indeed even the observation that they can causes increases or decreases, while interesting, is not novel (see Bolster, Le Borgne and

Dentz, Physics of Fluids 2009). - The BTCs in figure 5 are troubling, because it appears that they have different masses (e.g. the red continuous and dashed lines clearly do) – how is this possible with the given initial conditions? - Line 373 – this statement is either obvious by mass conservation or wrong because of what I note in my previous comment. - When I look at the Table 4, it looks like ADE is pretty darn good based on the error metrics the authors have chosen – $r^2$ is deceptive that way as it tells you very little about what's happening in tails and the likes (which I think is what the authors care about). If you really want to convey the message that the MIM is better another better chosen one would be in order. - It is clear from the figures that the MIM works better. While I am not surprised I really wish that the authors would make a strong effort to link the anomalous transport parameters to physical characteristics of the system – for example it is it the eddies that are causing the delays then you may be able to relate the delay time to a characteristic diffusion time (which you could test by running simulations with diffusion coefficients that span several orders of magnitude if needed). This would truly make this a paper worthy of publishing as these are the questions that we need answered in the anomalous transport community.

So, I am sorry to say that my recommendation for this paper is that it be rejected. There are solid elements to this work and I do believe that with some effort it will be publishable. However to reach the HESS standard I believe those efforts to be so substantial as to require much more time than is feasible for a standard revision.

---

## Referee Comment (RC2) · Anonymous Referee #2 · 23 Jul 2018

Summary: The authors study the impact of fracture roughness on non-Fickian transport through self-affine fractures. The roughness is decomposed into two different scales and authors show that the small-scale secondary roughness has major impact on non-Fickian transport. Finally, authors fit ADE and MIM and discuss how the secondary roughness affects the fitted model parameters.

General Comments: The idea of decomposing the fracture roughness into two different scales and studying its impact on transport is very interesting. However, the definition for the secondary roughness is too vague/qualitative. The level of analysis, interpretation and writing is not yet appropriate for HESS.

Major comments: 1. Overall writing should be improved: there are quite a few typos (some of them can be found in the specific comments), and a few parts lack details.

[Figure]

2. The sensitivity of the results with respect to the cut-off level should be discussed. It is not clear why different levels of approximation walls are chosen for the top and bottom fracture walls. It seems like some of the major results are caused by the fact that level 5 instead of level 4 was chosen for the bottom fracture wall.

3. Interpretation is not insightful enough: It is intuitive that the secondary roughness will make flow more heterogeneous, which in turn makes transport more non-Fickian. The authors do not provide enough insights for two different roughness scales. For example, can you quantitatively link the MIM model parameters with the two different roughness scales? Or can we have some understanding on the quantitative criteria for defining secondary roughness?

4. RTD is not properly defined.

5. It seems like fitted BTCs doesn't honor mass conservation (Fig 8). Please double check.

6. Table 4: $D_f/D_{Taylor}$ decreases as Peclet number increases. Can you explain why?

Specific Comments:

Line 25: "BTCs decreased" – increased??

Line 55-56: Not clear.

Line 69: "well-behaved BTCs" – vague

Line 121: MIM model is not inverse model.

Line 188: "level 4 and level 5" – What is "Zou's quantitative criterion"? The criteria for choosing the cut off should be discussed in detail.

Table 1: sigma is not defined.

Line 352-: More details on Table 2 should be provided.

Line 393: RTD not defined

Figure 8: Mass conservation?

Table 4: D_f/D_Taylor trend: can you explain?

Line 485: weak -> stronger?

Line 490: "curial" - typo